# Broadcasting American red squirrel vocalizations influences detection probability

**Ian G. Warkentin**[1]*, **Heather E. Spicer**[1], **Jenna P. B. McDermott**[2], **Darroch M. Whitaker**[3], **Erin E. Fraser**[1]

**1** Environmental Science Program, Memorial University of Newfoundland, Corner Brook, Canada,
**2** Cognitive and Behavioural Ecology Program, Memorial University of Newfoundland, St. John's, Canada,
**3** Parks Canada, Rocky Harbour, Canada

* ianw@mun.ca

## Abstract

Territorial responses by North American red squirrels (*Tamiasciurus hudsonicus*) to conspecifics vary seasonally with peaks during mating and dispersal periods. Broadcast of squirrel vocalizations during surveys may elicit territorial defense behaviors such as calling and movement that make individuals more available for detection, with implications for subsequent occupancy and abundance analyses. We examined the effect of vocalization broadcasts on detection probability during point counts throughout a 14-month period at two locations (year-round study) and during two summers at a third location (summer-only study) on Newfoundland, Canada. Overall detection probability based on sight and sound varied seasonally but the use of vocalization broadcast consistently enhanced detection probability. Squirrels were also more likely to be seen during vocalization broadcast survey periods than during silent point counts. Response to vocalization broadcast was highest when local population density was lowest. Higher detection probability during the initial silent periods of our surveys, when population density was high, likely reflects the greater chance of spontaneous vocalizations in response to the behavior of neighbors.

## Introduction

Our capacity to quantify population size or density and examine complex interactions between organisms and their environments has been greatly enhanced by increases in computational power and advances in statistical approaches [1,2]. Much of the field research conducted in these areas of investigation has its foundation in survey designs that depend upon the ability to detect the presence of the target organisms. However, the probability of detecting an individual during a survey is highly variable and has multiple components [3–5]: (1) the probability that the individual associated with a survey location is actually present at that location during the survey period (linked to organismal abundance and distribution), (2) the probability that the individual is available to be detected (either physically located or moving such that it could be seen by the surveyor, or vocalizing such that it could be heard by the surveyor) given that it is present, and (3) the probability that the individual will be detected by the surveyor (actually seen or heard) given that it is both present and available. Ultimately, while it is possible to calculate these different components of detectability [5–7]

**Data availability statement:** Data files are available through the Borealis data repository at https://doi.org/10.5683/SP3/FC8A9P.

**Funding:** Support for field research was provided by a Memorial University Environmental Science Research and Study Grant (HES) along with additional funding from the Center for Forestry Science and Innovation of the Government of Newfoundland and Labrador (DW, IW), Memorial University of Newfoundland (JPBM), Parks Canada (IW), and the Natural Sciences and Engineering Research Council of Canada (EF, IW). The funders had no role in study design, data collection and analysis, decision to publish, or preparation of the manuscript.

**Competing interests:** The authors have declared that no competing interests exist.

and to statistically account for imperfect detection [e.g., 8], statistical power and by extension ecological inference is far greater when the detection of individuals present in the landscape is effective and maximized [9,10].

For many highly territorial animals such as the North American red squirrel (*Tamiasciurus hudsonicus*; [11]), measurement of population size, distribution, or habitat occupancy typically has depended on detecting individuals through either live trapping or point counts with or without the broadcast (also referred to as playback) of conspecific vocalizations [12–14]. Red squirrels make highly visible and audible territorial displays during certain periods in the annual cycle [15] and so surveys based on detecting these displays (i.e., point counts relying on the squirrel being available for visual or auditory detection) can be effective. However, these territorial displays are intermittent, so broadcast of red squirrel vocalizations at point count locations has been used with the aim of eliciting behavioral responses from territorial individuals that would not otherwise signal their presence during a silent point count (i.e., increasing the availability component of detection probability). In theory, vocalization broadcasts simulate the presence of an intruder and provoke vocal or other signalling behavior from territory holders [11], but the effectiveness of this technique for increasing detection probability in red squirrels has received limited study.

Chavel et al. (2017) [16] compared red squirrel detection probability among three survey methods: trapping, silent point counts, and point counts using conspecific vocalization broadcasts. They reported that the detection probability from early June through mid-August was somewhat lower when using point counts with vocalization broadcast, but not statistically different from those for either live trapping or silent point counts when examined using dynamic occupancy modelling approaches. The absence of an effect of vocalization broadcast on the detection probability of squirrels is somewhat surprising given its common use for surveys. Their sampling period also was restricted to one summer, and so they could not assess whether there are any temporal (seasonal or annual) patterns in the potential improvement of detectability associated with using vocalization broadcast.

We build on the work of Chavel et al. (2017) [16] by investigating and characterizing the role of vocalization broadcasts in increasing detection probability of red squirrels during point count surveys throughout the annual cycle and under varying demographic circumstances. Our first objective was to assess whether there were differences in detection probability between silent point counts and point counts that included vocalization broadcasts. Given the theoretical expectations outlined above [11], we predicted that vocalization broadcast would lead to an improved probability of detection for red squirrels over that for silent point counts. Our second objective was to determine whether any improvement in probability of detection based on the use of vocalization broadcast varied across seasons. Red squirrels maintain solitary territories that they defend against other squirrels regardless of their sex [12]; the only exception to this occurs during parental care when females care for pups from birth through juvenile independence [14]. Red squirrels vary in their use of visible and vocal behavior, and potentially their response to conspecifics, across seasons [11,13,15] with increased activity during periods when territorial intrusions are more frequent, such as during the reproductive season and fall dispersal [17]. Consequently, we predicted that squirrels would be more responsive to vocalization broadcasts during mating (March-April) and dispersal (August-September) periods, when territorial intrusions were more likely to occur [13], thus increasing the availability component of detection probability. We further predicted that vocalization broadcasts would improve the quality of squirrel detection information by encouraging squirrels to approach the point count site (following [17]), potentially allowing researchers to visually collect data from individuals. Our third objective was to assess the importance of population density (where number of detections was used as a proxy for

density) on squirrel response to vocalization broadcasts. As density may influence the rate of territorial intrusions [18], we predicted that detection rate would be higher in response to vocalization broadcasts when squirrel population densities were higher, as occurs for example when cone crops reach their peak [19].

## Materials and methods

### Study areas

We used data from two serendipitously co-occurring studies conducted at three sites on Newfoundland, Canada. Red squirrels were introduced successfully to this ~ 108,000 km² boreal island during the 1960s and had colonized most suitable habitat by ~ 1995 [20]. The first study was a year-round project which sampled red squirrels in forested suburban parks from August 2016 through September 2017 in the communities of Corner Brook (48° 57′ N, 57° 56′ E; 50 to 130 m elevation at survey points) and Pasadena (48° 59′ N, 57° 35' E; 70 to 170 m elevation). The second study was a summer-only project which sampled red squirrels and forest birds across a 257 km² portion of the Upper Humber River and Main River watersheds of the Long Range Mountains of western Newfoundland (centered 77 km north of Pasadena at 49° 40´N, 57° 16´ E; 75 to 608 m elevation; see [21] for detailed description of this study area) from early June through mid-July of 2016 and 2017. Forests in all study areas are representative of the eastern Canadian boreal forest and are referred to respectively as the Corner Brook and Northern Peninsula Forest Regions [22,23]. Described as "wet balsam fir forests" [24], these areas are dominated by even-aged mixed coniferous and deciduous species stands as well as single-species forest stands dominated by balsam fir (*Abies balsamea*) or, less often, black spruce (*Picea mariana*) within a broader landscape matrix that includes extensive bogs, barrens and other natural openings [25].

### Field methods

For the year-round project, we conducted counts at eight points along an established trail in each of the two communities (the Ginger Route in Corner Brook and a trail in the Pasadena Ski and Nature Park in Pasadena). Points were at least 300 m apart to avoid detecting the same individual at adjacent locations [13,26,38], with point counts conducted during the third week of each month. Surveys began two hours after sunrise and continued for three to four hours until all point counts were complete. Point count order of visitation was reversed in alternating months and surveys were not conducted during heavy precipitation (rain or snow) or high winds (Beaufort Wind Scale Force 5; > 29 km/h). Visits to points consisted of an eight-minute silent listening period followed immediately by an eight-minute vocalization broadcast period consisting of 14-second segments of recorded red squirrel territorial calls (Macaulay Library, Cornell Lab of Ornithology; Catalogue numbers: 100916, 136185) followed by 10 seconds of silence repeated five times during each of the four 2-minute intervals making up the 8-minute broadcast period. We broadcast vocalizations using a JBL Flip 3 portable Bluetooth speaker (Harman International Industries, Incorporated, Northridge, CA 91329 USA) which produced an average volume of 80 dB at 1 m from the speaker. During each full 16-minute point count (silent + broadcast periods), the observer (all surveys completed by HES) recorded all detections of individual red squirrels, as well as the type of detection (i.e., seen, heard, or both seen and heard). We categorized observations as visual if the squirrel was seen only or both seen and heard during the sampling period; and auditory if the squirrel was heard only.

We quantified red squirrel activity in two ways for each point count. First, individual squirrels were recorded upon initial visual or auditory observation in each of the point count periods (i.e., an individual squirrel detected during the silent period would be counted again if

it was still evident and detectable during the vocalization broadcast period). The type of detection associated with an individual squirrel also could change between periods; for example, an individual could be counted as an auditory detection during the silent period and as a visual detection during the vocalization broadcast period. Second, we amalgamated the two point-count periods into one total 16-minute point count period (silent + broadcast) and created a tally of all individuals detected regardless of the period during which the detection occurred.

Surveys were allocated to one of four seasonal periods, based on known seasonality of red squirrel behavior [11–13,27,28] and the climate of the study area. Data collected during August-September were considered to come from "early autumn", when young of the year and some adults may be dispersing to new territories. October through December was considered "late autumn", when squirrels were assumed to be established in their territories. January through April was "winter" and corresponded with the time period in western Newfoundland when there is predictable snow cover. May through July was "spring/summer" and corresponded with the pup-rearing period for red squirrels, with mating assumed to have taken place during late winter [28]. Because the survey protocol spanned 14 months, the analyses included two "early autumn" seasons.

For the summer-only project, sampling locations were placed in a grid of points spaced 500 m apart across the study area, with a total of 992 points visited once each during 33 days of sampling in 2016 (JPBM and three other observers). The survey grid was shifted 250 m north and east for the 2017 season, meaning that points for the second year of sampling fell midway between the points sampled in 2016, and 969 points were each visited once during 27 days of sampling (JPBM and four other observers). Each visit consisted of five periods (11 total minutes) which we separated in the following fashion: three two-minute periods of silent listening, followed by a period with a two minute broadcast of gray-cheeked thrush (*Catharus minimus*) song and calls (the target of related research using this protocol, see [29]; FoxPro model FX3 or Crossfire game callers; FoxPro Incorporated, Lewistown, PA 17044, USA) followed by one minute of silence, and finally a fifth period with one minute of red squirrel vocalization broadcast using the same recording as the year-round project and including one final minute of silence. The volume control of broadcasting units was set at a constant level for all surveys, and when measured 1 m from the speaker the average volume of red squirrel vocalizations was 53.7 dB. Regardless of mode of detection (visual versus auditory was not recorded), only the initial period of detection was noted and detections of the same individual during subsequent periods were not included in this analysis.

For both projects, we used the total number of squirrels detected in a standardized time period (our point-count visit) as a proxy for population density. While there are issues with this approach [30], we used these proxy measures only to make comparisons between replicated survey efforts (i.e., the results of the early autumn surveys for the year-round project, and the results of the annual surveys for the summer-only project), reducing the likelihood that variation in season and space may have affected our ability to detect squirrels. Squirrels were not marked for either project; however, movements by individuals detected during a point count were monitored as closely as possible in an effort to avoid double counting. All work was completed with the appropriate permits (Newfoundland and Labrador provincial scientific research permits WLR2016-18, WLR2017-15 and ESA2016/17-05) and protocols were approved by the Institutional Animal Care Committee of Memorial University of Newfoundland (16-05-EF, 16-16-IW).

## Statistical analyses

For the year-round project, we created two sets of competing multi-season occupancy models [31] to examine the value of using vocalization broadcasts, as well as to assess differences in the detection probability of red squirrels among seasons and between the two sites included in

the year-round dataset. Each model set included a null model that did not include any explanatory variables and which was used as a reference to assess the explanatory power of the other models. Analyses were conducted using the software package PRESENCE (version 12.12; [32]). We applied the first model set to the dataset to compare the effectiveness of silence and listening vs. vocalization broadcasts and listening at point counts. We used data collected during the eight-minute silent periods and vocalization broadcast periods, where squirrels that had previously been detected in the silent period were considered new detections if they were subsequently detected in the broadcast period. We included an interaction term (season*method), which enabled us to assess whether any improvement in detection probability from vocalization broadcast varied among seasons. We applied the second model set to further assess the value of including both silent and vocalization broadcast components in point count surveys using data from the complete 16-minute observation period, but only incorporated the first detection of each squirrel. For each analysis the data were fit to simple, multi-season occupancy models that estimated initial site occupancy ($\psi$), along with seasonally varying colonization ($\gamma$) and extinction ($\delta$) rates as they influenced the probability of species detection at a site ($p$) on a seasonal (model sets 1 and 2) and sampling method (model set 1) basis. We ranked competing models based on Akaike's Information Criterion (AIC), a measure of model likelihood, where the lowest AIC value indicates the most parsimonious model [33]. Support for each model was reflected by the Akaike weight, which reflects the likelihood that a given model is the best model and is calculated based on the difference between the AIC value of a given model and that of the most parsimonious model ($\Delta$AIC). To compare the types of squirrel detections (visual versus auditory) achieved during the silent and vocalization broadcast periods, we used a Chi-squared test to compare the proportion of total detections that were visual (as compared to auditory) in the silent vs. vocalization broadcast periods.

For the summer-only project we followed Powell et al. (2014) [34] and treated each of the five time periods per point count as separate visits in a single-season occupancy modeling analysis using the software package PRESENCE (as described above). This approach enabled us to compare detection probability across periods and consequently assess differences in detection probability between silent and vocalization broadcast components of the visit. Because point count locations were moved each year and the annual datasets effectively constituted separate samples, we did not consider multi-season modeling approaches. To model occupancy for each year, we built models with all possible combinations of the six habitat variables identified which resulted in a "best-fit" model during related research (see [21]). Despite a large inter-annual difference in our proxy for squirrel population density in this study area, there were only slight differences between years in the components of the best-fit habitat model of squirrel occupancy. Common elements across years in the best-fit habitat model were elevation as well as the extent of conifer scrub and balsam fir-black spruce stands of intermediate age (30–70 years old). For 2016, extent of coverage by water bodies and balsam fir-black spruce stands of younger age classes (10–30 years old) were included with the three variables listed above; while for 2017 the best-fit model included the extent of open habitat along with the three common variables in the initial list. This best fit habitat model was used as a base or null model, to which additional terms were added to evaluate the influence of silent and broadcast periods on detection probability. For each year of data collection, we modeled detection probability across (1) all five survey periods separately (labelled as broadcast_5 in models), and an aggregation of these five segments into (2) three periods that separated the initial two-minute silent period from the middle three elements (the two subsequent two-minute silent periods along with the thrush broadcast period) and the fifth squirrel vocalization broadcast period (broadcast_3), or (3) two periods that separated the initial silent period from a combination of the latter four elements (two two-minute silent periods plus the

thrush broadcast period and the squirrel broadcast period (broadcast_2), or (4) two periods that combined the initial four elements (three two-minute silent periods plus the thrush broadcast period) into a single period distinct from the fifth squirrel broadcast period (broadcast_1). Each model set included the null model based on habitat variables described above which was used to assess the explanatory power of the other models.

## Results

### Year-round study

Across 14 months of repeated sampling at 16 points during the year-round study (a total of 224 point counts), there were 148 unique (within 16-minute point count duration; Table 1) squirrel detections (mean ± $SD$ of 0.67 ± 0.76 detections per point count visit). Within seasons, detections ranged from a low during the winter period of 17 detections over 4 months (0.27 ± 0.51 detections per point count visit) to a high of 48 detections during August and September (early autumn 2017; 1.50 ± 0.80 detections per point count visit). When we included all within-period detections of squirrels (i.e., the same individual may have been noted during one or both periods of the point count, the latter regarded as two detections), total detections increased to 212 (mean of 0.95 ± 1.23 detections per point count visit; 24 detections during winter for 0.38 ± 0.75 detections per point count visit and 77 detections in early autumn 2017 for 2.41 ± 1.60 detections per point count visit). As reflected by the number of unique detections during each two-month early autumn season, the presumed local population densities of red squirrels were lower during early autumn 2016 (n = 31) compared to early autumn 2017 (n = 48).

The proportion of visual versus auditory detections was significantly lower during the silent period of point counts (19%, 16 of 86) compared to the proportion of visual detections during vocalization broadcast (35%, 44 of 126; χ2 = 6.706, $d.f.$ = 1, $P$ = 0.009). With the exception of one point count in August 2016 and one in August 2017 (two visual detections each), no more than one squirrel was ever visually detected at an individual point count. Visual detections varied seasonally; half of the broadcast period visual detections (22 of 44) occurred during the early autumn of both years combined and a further 25% (11 of 44) occurred during late autumn (Table 1). In total, we noted 11 squirrels initially counted as auditory detections during a silent period that were then recorded as visual detections during the associated vocalization broadcast period. By comparison, only four individuals that we noted as visual detections during a silent period were then subsequently recorded as auditory detections during the vocalization broadcast.

Based on multi-season occupancy modelling, the most parsimonious model from model set 1 included season and sampling method (silent vs. vocalization broadcast) as the only variables influencing detection probability (Table 2). For model set 2, which in effect ignores

**Table 1. Detections of American red squirrels during point count visits of year-round study.** Values are based on season of sampling, phase of point count (silent versus call broadcast periods), and mode of detection (number of detections and percentage which were visual).

| | Early Autumn 2016 | Late Autumn 2016 | Winter 2017 | Spring-Summer 2017 | Early Autumn 2017 | Total |
|---|---|---|---|---|---|---|
| Point counts conducted | 32 | 48 | 64 | 48 | 32 | 224 |
| Silent period detections (% visual) | 21 (9.5) | 13 (7.7) | 12 (33.3) | 7 (28.5) | 33 (21.2) | 86 (18.6) |
| Vocalization broadcast detections (% visual) | 25 (32.0) | 27 (40.7) | 12 (50.0) | 18 (38.9) | 44 (31.8) | 126 (34.9) |
| Total detections (average per point count) | 46 (1.4) | 40 (0.8) | 24 (0.4) | 25 (0.5) | 77 (2.4) | 212 |

**Table 2. Multi-season occupancy model selection results.** Performance of *a-priori* models to describe the influence of season and sampling site (model sets 1 and 2) and sampling method (silent vs. call broadcast; model set 1 only) on probability of detection (*p*) of American red squirrels during monthly point count surveys over 14 months in western Newfoundland. The null model does not include any explanatory variables and is used to assess the explanatory power of the competing models.

| Model | AIC | ΔAIC | Weight | Number of parameters |
|---|---|---|---|---|
| *Model set 1* | | | | |
| *p* (season, method) | 500.82 | 0 | 0.7698 | 16 |
| *p* (season, method, site) | 503.26 | 2.44 | 0.2273 | 18 |
| *p* (season) | 512.57 | 11.75 | 0.0022 | 14 |
| *p* (season, site) | 515.04 | 14.19 | 0.0006 | 16 |
| *p* (season, method, season*method) | 518.40 | 17.58 | 0.0001 | 26 |
| *p* (method) | 522.60 | 21.78 | 0 | 11 |
| *p* (method, site) | 525.62 | 24.8 | 0 | 13 |
| *p* (null) | 534.99 | 34.17 | 0 | 10 |
| *p* (site) | 536.05 | 35.23 | 0 | 11 |
| *Model set 2* | | | | |
| *p* (season) | 267.30 | 0.00 | 0.8259 | 14 |
| *p* (season, site) | 270.44 | 3.14 | 0.1718 | 16 |
| *null* | 279.79 | 12.49 | 0.0016 | 10 |
| *p* (site) | 281.61 | 14.31 | 0.0006 | 11 |

sampling method, the most parsimonious model included season only (Table 2). Site (Corner Brook versus Pasadena) was not important (ΔAIC > 2) when modeled alone or as a co-variate in either model set. We estimated all detection probability results for the year-round project using the best-fit models from each model set.

We identified substantial variation in detection probability among seasons and between sampling methods (Fig 1). Regardless of sampling method, overall probability of detection was greatest during early autumn and lowest during winter and spring/summer. Across seasons, the silent sampling method resulted in the lowest probability of detection, with estimates of detection probability that were between 17 and 22% lower than when call broadcast was used. The best silent + vocalization broadcast model had only slightly higher (12%) or approximately equal (1% lower) probability of detection estimates for each season than did the best vocalization broadcast models. Probability of detection increased between 11 and 34% (winter and late autumn, respectively) when a vocalization broadcast was added to a silent count.

## Summer-only study

During the summer-only project we detected a total of 241 squirrels from 992 points sampled in 2016 (0.24 ± 0.56 detections per point), with a maximum of 4 individuals detected at one point. By contrast, during 2017 only 47 squirrel detections were noted from 969 points (0.05 ± 0.21 detections per point), with a maximum of two individuals detected at one point. The latter detection rate is only 10% of that for the simultaneously collected spring/summer season data from the year-round project. A significantly lower proportion of initial squirrel detections (14%, 35 of 241) occurred during the broadcast portion of the point counts conducted in 2016 when compared to the proportion first detected during the broadcast portion of point counts conducted in 2017 (40%, 19 of 47; χ2 = 17.326, *d.f.* = 2, *P* < 0.001; Fig 2). Total detections of red squirrels during the broadcast of gray-cheeked thrush vocalizations were similar between years (16% [38 of 241] and 11% [5 of 47]; respectively).

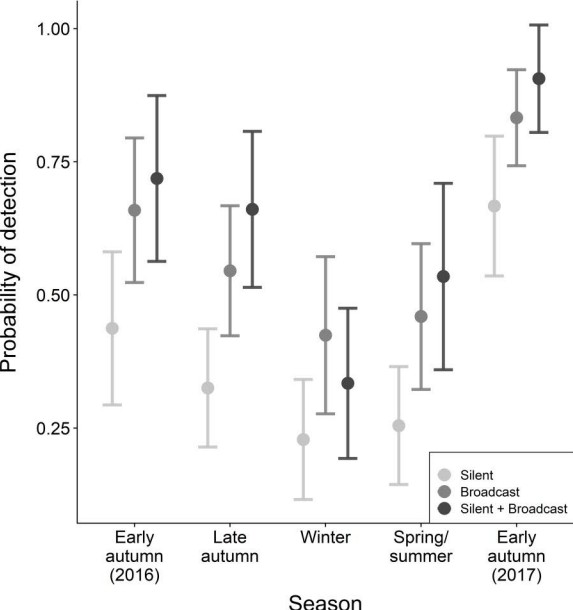

**Fig 1. Estimates of seasonal probability of detection ( ± 95% confidence interval) of American red squirrels from year-round study in western Newfoundland, Canada.** For August 2016 through September 2017, detection probabilities were estimated using a multi-season occupancy model applied to datasets that included observations from the silent portion of point counts, observations from the portion of point counts using broadcasts of squirrel vocalizations, or the combined observations from both silent and broadcast periods.

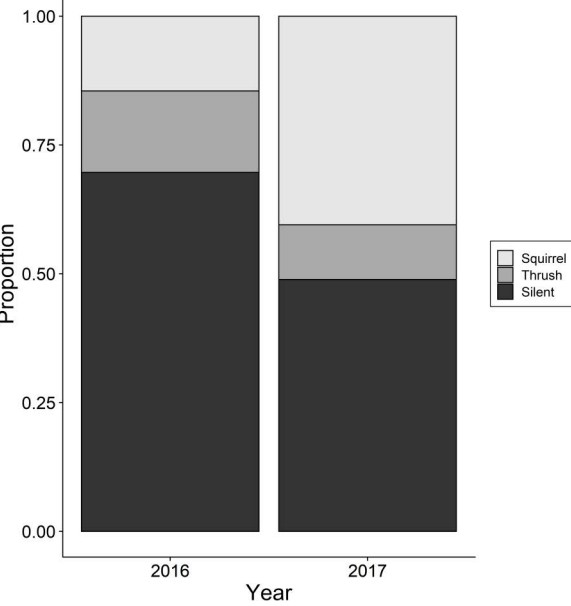

**Fig 2. Proportion of initial detections for point count periods.** Point count detections (during silent, gray-cheeked thrush broadcast, and red squirrel broadcast) for summer-only season from 2016 (n = 241 detections during 992 point surveys) and 2017 (n = 47 detections during 969 point surveys).

As expected from the distribution of squirrel detections across point count periods noted above, the best-fit models ($\Delta$AIC < 2) for detection probability based on single-season analysis for each year included either the initial silent period (broadcast_2 and broadcast_3 for 2016, Table 3) or the squirrel vocalization broadcast period (broadcast_3 and broadcast_1 for 2017, Table 3). Detection probabilities ($\pm SE$) during each assessment period of the broadcast_3 models for 2016 versus 2017 were: 0.2351 ± 0.0269 versus 0.0457 ± 0.0181 (first silent period), 0.1355 ± 0.0131 versus 0.0221 ± 0.0080 (second period combining 2 two-minute silent segments and the gray-cheeked thrush broadcast), and 0.1108 ± 0.0189 versus 0.0789 ± 0.0272 (final squirrel broadcast segment).

## Discussion

Our results highlight the value of vocalization broadcasts for improving detection probability of red squirrels compared to silent point counts, although the influence varies with circumstances. As per our prediction, the use of vocalization broadcast did improve the probability of detecting red squirrels. Of the three components of detection probability [3–5], using vocalization broadcast during red squirrel point-count surveys likely most influenced the probability of an individual being available to be detected. American red squirrels engage in limited movements during much of the annual cycle [28] and their territories frequently measure less than 1 ha [28,35]. Consequently, if a point count location is near, or on, an occupied territory, then squirrel presence near that location is probable both during silent point counts and broadcast vocalizations (i.e., off-territory forays are limited for much of the year but do occur in both males and females; [36–38]). Thus, the most likely mechanism for vocalization broadcast to enhance the potential for an observer to detect a squirrel is through increased movements or vocalizations by squirrels in response to that broadcast. Red squirrels are known to respond with greater intensity (i.e., through both more frequent vocalizations but also more movement) to unfamiliar calls than to those from familiar neighbors [11]. Our use of vocalization broadcast based on calls from outside the population (our recordings were from Quebec and Nova Scotia, Canada) took advantage of this behavior and presumably would elicit a higher rate of response, though we did not test this using locally recorded vocalizations.

**Table 3. Single-season occupancy model selection results.** Performance of *a-priori* models to describe the influence of vocalization broadcast on probability of detection (*p*) of American red squirrel during point count surveys during the summers of 2016 and 2017 in western Newfoundland. All models were based on an optimal model for site occupancy ($\psi$) that incorporated elevation and extents of conifer scrub, open habitat and water within the survey area [21]. The null model only includes habitat variables influencing occupancy and was used to assess the explanatory power of the competing models regarding detectability.

| Model | AIC | $\Delta$AIC | Weight | Number of parameters |
|---|---|---|---|---|
| *Models for summer 2016* | | | | |
| *p* (broadcast_2) | 1471.46 | 0.00 | 0.5478 | 7 |
| *p* (broadcast_3) | 1472.22 | 0.76 | 0.3746 | 8 |
| *p* (broadcast_5) | 1475.39 | 3.93 | 0.0768 | 10 |
| *p* (broadcast_1) | 1485.61 | 14.15 | 0.0005 | 7 |
| *Null* | 1488.47 | 17.01 | 0.0001 | 6 |
| *Models for summer 2017* | | | | |
| *p* (broadcast_3) | 476.15 | 0.00 | 0.5851 | 7 |
| *p* (broadcast_1) | 477.46 | 1.31 | 0.3039 | 6 |
| *p* (broadcast_5) | 480.03 | 3.88 | 0.0841 | 9 |
| *Null* | 486.82 | 10.67 | 0.0028 | 5 |
| *p* (broadcast_2) | 488.38 | 12.23 | 0.0013 | 6 |

Overall detection probability varied substantially among seasons, but contrary to our prediction, the extent of improvement in detection provided by vocalization broadcast was relatively consistent across seasons at around 20%. Our results suggest that the addition of vocalization broadcasts increases detection probability of red squirrels, likely through increases in availability (given presence) throughout the annual cycle. Seasonal changes in overall detection probability can result from both seasonal shifts in squirrel occupancy and/or detectability. Our study design does not allow us to separate these independent factors, but we suggest that the drop in number of detections and detection probability during the winter period reflects changes in detectability rather than occupancy. Previous work has demonstrated that squirrel activity levels decline with decreasing seasonal temperature and in association with life history strategies that revolve around energy and food store conservation [28,39,40] as opposed to movement away from breeding areas. In fact, red squirrels are known to defend territories from intruders throughout the year [11] and territory resources are particularly important to overwinter survival [41].

The highest seasonal detection probability during the year-round study was from early through late autumn, which may reflect an influx of new squirrels to the study site, or a seasonal change in squirrel behaviour (and thus detectability). Dispersal of yearlings and adults in autumn [12,26] likely increases both the number of squirrels present and the need for territory defense and, consequently, responses to the calls of other potentially intruding conspecifics [42]. This is also a period when territorial individuals are building and defending their food stores for the coming winter and consequently competition and conflict may be highest [28].

Not only did vocalization broadcasts improve detection probability in comparison to silent point counts, but it also appeared that using vocalization broadcast alone with no accompanying silent point count may be sufficient in some situations. The difference in detection probability between an 8-minute vocalization broadcast only and a 16-minute silent point count + vocalization broadcast was non-existent during most seasons (Fig 1). The advantage of doing a full 16-minute count including silent + vocalization broadcast was most evident during late autumn. While our results point to a clear benefit of vocalization broadcasts over silent point counts, previous work suggests that these benefits may not be consistent among locations. Unlike the pattern that we observed, Chavel et al. (2017) [16] did not identify significant differences in detection probability between silent point counts and those using vocalization broadcast. While detection probability may not be different between these two approaches in all studies or in all seasons or years, there may still be benefits to using vocalization broadcast if it increases the quality of the detection or reduces temporal variation in the probability of detection. For most passive studies (i.e., those not trapping squirrels) the majority of field time and costs is spent accessing survey points. Consequently, the relatively minor increase in time and cost of using the composite silent + broadcast technique may be worthwhile even if increases in detection probability are only slight or uncertain.

The results of our year-round study support the use of vocalization broadcast over silent point counts to increase the quality of the detections made, as the proportion of visual detections was much higher during vocalization broadcast than silent periods, particularly during early through late autumn. Visual detections may be preferred when the study design requires researchers to collect data about known/marked individuals. Increased levels of visual rather than auditory detection associated with vocalization broadcast have also been reported for songbird surveys [43], with the advantage being an ability to identify individual markers or detect behavior indicative of specific phases of the breeding cycle (e.g., [44]).

We did note an apparent influence of population density on responses for the summer-only study, which contrasts the findings of Shonfield et al. (2012) [18]. Although we had no direct measure of population density for our study areas, we used the number of detections as a

proxy for relative density to compare values across sample periods/years for the same locations (analogous to [45,46]; although see [47]) with a particularly large standardized sample for the summer-only study. For the summer-only project, 39% of first detections occurred during the initial silent period of 2016 (when population density was high) versus 25% of first detections during that same period of point count visits in 2017 (Fig 2). By contrast, in 2016 14% of first detections occurred during the squirrel broadcast period of the visit and consequently appeared to be elicited by that broadcast, whereas in 2017, when squirrel population density was much lower, 40% of first detections occurred during the squirrel broadcasts. It appears that red squirrels are more likely to spontaneously vocalize, or be detected moving about their territories, when population densities are high (i.e., availability is greater). Thus, our findings suggest that broadcasting vocalizations may be particularly effective at lower densities when there are fewer potential intruders to provoke squirrels to spontaneously vocalize during surveys conducted in the absence of vocalization broadcast.

The use of vocalization broadcast to enhance detection probability during surveys is common across a range of mammals (e.g., canids: [48,49]; primates: [50,51]; marine mammals: [52]; ground squirrels: [53]; African carnivores: [54,55]). Clearly the effectiveness of this technique is context specific for various taxa (e.g., see [56] regarding lions [*Panthera leo*] and spotted hyaenas [*Crocuta crocuta*] or [11,13,18] regarding red squirrels). Our findings indicate that, for red squirrels, not only is detection probability enhanced by using vocalization broadcasts to elicit responses, but also that this response is consistent throughout the year, although likely diminished as population density increases. Enhanced detection probability (i.e., availability) in combination with an ability to statistically account for imperfect detection [8] will increase survey efficacy; vocalization broadcast is a useful tool in this process.

## Acknowledgments

We thank the field assistants and colleagues who supported this research.

## Author contributions

**Conceptualization:** Ian G. Warkentin, Heather E. Spicer, Jenna P. B. McDermott, Darroch M. Whitaker, Erin E. Fraser.

**Data curation:** Heather E. Spicer, Jenna P. B. McDermott.

**Formal analysis:** Ian G. Warkentin, Heather E. Spicer, Jenna P. B. McDermott, Erin E. Fraser.

**Funding acquisition:** Ian G. Warkentin, Darroch M. Whitaker, Erin E. Fraser.

**Investigation:** Heather E. Spicer, Jenna P. B. McDermott.

**Project administration:** Ian G. Warkentin, Darroch M. Whitaker, Erin E. Fraser.

**Resources:** Darroch M. Whitaker, Erin E. Fraser.

**Supervision:** Ian G. Warkentin, Darroch M. Whitaker, Erin E. Fraser.

**Writing – original draft:** Ian G. Warkentin.

**Writing – review & editing:** Ian G. Warkentin, Heather E. Spicer, Jenna P. B. McDermott, Darroch M. Whitaker, Erin E. Fraser.

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
