## [Decision Letter · Decision Letter 0]

27 Dec 2024

PONE-D-24-56431Broadcasting American red squirrel vocalizations influences detection probabilityPLOS ONE

Dear Dr. Warkentin,

Thank you for submitting your manuscript to PLOS ONE. After careful consideration, we feel that it has merit but does not fully meet PLOS ONE’s publication criteria as it currently stands. Therefore, we invite you to submit a revised version of the manuscript that addresses the points raised during the review process.

We look forward to receiving your revised manuscript.

Kind regards,

Gisele Akemi Oda, Ph.D.

Academic Editor

PLOS ONE

Journal Requirements:

“Support for field research was provided by a Memorial University Environmental Science Research and Study Grant (HES) along with additional funding from the Center for Forestry Science and Innovation of the Government of Newfoundland and Labrador (DW, IW), Memorial University of Newfoundland (JPBM), Parks Canada (IW), and the Natural Sciences and Engineering Research Council of Canada (EF, IW). None of the funding agencies played a role in the development, conduct, or interpretation of our findings. “

3. In the online submission form, you indicated that your data will be submitted to a repository upon acceptance.  We strongly recommend all authors deposit their data before acceptance, as the process can be lengthy and hold up publication timelines. Please note that, though access restrictions are acceptable now, your entire minimal  dataset will need to be made freely accessible if your manuscript is accepted for publication. This policy applies to all data except where public deposition would breach compliance with the protocol approved by your research ethics board. If you are unable to adhere to our open data policy, please kindly revise your statement to explain your reasoning and we will seek the editor's input on an exemption.

Reviewers' comments:

Reviewer's Responses to Questions

**Comments to the Author**

1. Is the manuscript technically sound, and do the data support the conclusions?

Reviewer #1: Partly

Reviewer #2: Yes

2. Has the statistical analysis been performed appropriately and rigorously? 

Reviewer #1: N/A

Reviewer #2: Yes

3. Have the authors made all data underlying the findings in their manuscript fully available?

Reviewer #1: Yes

Reviewer #2: Yes

4. Is the manuscript presented in an intelligible fashion and written in standard English?

Reviewer #1: Yes

Reviewer #2: Yes

5. Review Comments to the Author

Reviewer #1: The study is interesting, but my concern is if this is a 'crucial' contribution to the census methodology you can use, when results seem to be dependent on the zone sampled and on the density of animals, that is one of the data you want to know. My other concern is that sound broadcasts could be provoking behavioral changes in the population that could alter your sampling. Maybe I am an old fashioned guy, but I find more reliable the capture, marking and releasing method for assesing this kind of population data.

Reviewer #2: The study was very well planned and conducted. I have some minor aspects to comment about.

Methodological aspects:

Would you consider including information about the species social life in the introduction? It has a methodological relevance. I guess they live in isolation; if animals live in pairs, how would you know who reacted to the broadcast?

Line 117-118 you stablished sampling points at least 300 m apart to avoid detecting the same individual at adjacent locations based in [13,26]. But [13] found that "prior to settlement, offspring made forays of up to 900 m off the natal territory." and "the farthest settlement distance was only 323 m from the natal territory." For that I understand that a young individual in the dispersal phase could eventually be counted twice. Would they also react to broadcats? Could you distinguish young and adult animals?

Line 120-122: how was that reversion of point count order in alternating months? Once you used the numerical crescent order and then the descrescent order? And what happened when surveys could not be conducted due to precipitation or winds, did you go back to survey some days later?

Minor corrections:

Line 42: delete space between "abundance and distribution"

Note that some registries in the refence list ends with a "." and others not.

6. PLOS authors have the option to publish the peer review history of their article (what does this mean? ). If published, this will include your full peer review and any attached files.

**Do you want your identity to be public for this peer review?** For information about this choice, including consent withdrawal, please see our Privacy Policy .

Reviewer #1: No

Reviewer #2: **Yes: ** Patrícia F. Monticelli

---

## [Author Response · Author response to Decision Letter 1]

24 Jan 2025

Comments to the Author

5. Review Comments to the Author

Reviewer #1: The study is interesting, but my concern is if this is a 'crucial' contribution to the census methodology you can use, when results seem to be dependent on the zone sampled and on the density of animals, that is one of the data you want to know.

Response: There are multiple tools available for censusing wildlife populations, among which counts of individuals at points or along transects are common approaches. As an enhancement to silent listening/watching at a location, some researchers have employed the broadcast of vocalizations to increase their rates of detection for individuals in surveyed populations. With any survey technique devised to assess wildlife population size, habitat structure (both at point count/transect locations and across the study area) along with population characteristics, will influence (1) the presence of individuals at a sample location, (2) how available individuals are for being detected, and (3) the capacity for the surveyor to detect those individuals present. There has been only one published study that we are aware of which examined whether the use of vocalization broadcast does improve detectability, and the results from that study were inconclusive (see manuscript for citation). Our objective was to test whether the use of vocalization broadcast might improve the detection rate of American red squirrels compared with simple silent listening at point counts in our study area. We also endeavoured to provide potential explanations for the variability in responses that we observed. Our findings point to circumstances that must be considered when devising survey protocols and interpreting their outcomes.

Recent statistical advances have also greatly enhanced the ability of researchers to account for variation in the detection probability of individuals over time and space, and we presume / expect that practitioners would employ these occupancy models in their analyses as we have done here

Reviewer #1: My other concern is that sound broadcasts could be provoking behavioral changes in the population that could alter your sampling. Maybe I am an old fashioned guy, but I find more reliable the capture, marking and releasing method for assessing this kind of population data.

Response: During our summer-only protocol, squirrels at a point count location were exposed to red squirrel vocalization broadcast for a 1-minute period once per summer. With 500 m between sample points within years, and an average 40 minute interval between broadcasts at adjacent points, there is a very low probability that an individual squirrel would be exposed to the broadcast from adjacent points multiple times in the same year. There was the potential for some individuals to experience this exposure again during the second year (if they survived between years) from point counts conducted at a 354 m diagonal distance during 2017. We find it difficult to equate this exposure to the broadcasts with a provocation/stimulus that could lead to extensive behavioural changes by squirrels, or that would alter our ability to determine and interpret detectability within and between years for this part of the study.

For the year-round study, point count locations were revisited on a monthly basis for 14 months, meaning that it is quite likely that individuals who survived through part or all of this timeframe were exposed to vocalization broadcast on multiple occasions. At a monthly interval, squirrels present experienced an eight-minute vocalization broadcast period consisting of 14-second segments of recorded red squirrel territorial calls followed by 10 seconds of silence repeated five times during each of the four 2-minute intervals making up the 8-minute broadcast period. As is evident from the data presented in Figure 1 of our manuscript, there was a decreased probability of detection from early autumn 2016 through winter which could potentially be attributed to modified behaviour in response to repeated exposure to the vocalization broadcast. However, this interpretation seems unlikely as there is a subsequent increase in probability of detection from winter through to the following early autumn 2017. Such a seasonal pattern is hard to reconcile with the hypothesis that the use of vocalization broadcasts has either negatively or positively altered the behaviour of red squirrels in our study population with respect to their presence at the point count locations or their responses to the vocalization broadcasts.

It is also worth noting that red squirrels have been documented as typically experiencing a high frequency of calls and rattles (territorial vocalizations) under natural circumstances. Dantzer et al. (Dantzer, B., Boutin, S., Humphries, M. M., & McAdam, A. G. (2012). Behavioral responses of territorial red squirrels to natural and experimental variation in population density. Behavioral Ecology and Sociobiology, 66(6), 865–878. https://doi.org/10.1007/S0026 5-012-1335-2) recorded a mean frequency of rattles by their study squirrels at slightly less than one rattle every seven minutes during daylight hours. From the perspective of a red squirrel, hearing territorial calls by conspecifics is likely a regular occurrence.

All researcher interactions with wildlife have the potential to influence the behaviour of the individuals being studied. Those interactions involving physical contact (particularly through capture and marking), may have a higher probability of altering behaviour compared to brief, largely non-invasive exposure to vocalization broadcasts. Chavel et al. (2017) found that the detection probability of squirrels from point counts was no different from that associated with live-trapping, and pointed to the value of using point count methodology when measuring American red squirrel occupancy. Price et al. (2009) carried out experiments on red squirrels with extensive use of vocalization broadcast, over multiple days in succession for some aspects of their protocol, on 51 individuals. They did not report any abandonment of territory or behaviour that was unexpected based on their theoretical predictions. Note that citations for the papers above are included in the manuscript.

Ultimately, it is up to individual researchers to compare survey methods and select those that they deem most appropriate given the goals of their research. We feel that the approach that we have presented here, which can enhance detection rates, is cost and time efficient, and is non-invasive and poses negligible risk to squirrels, may be of great value in many studies.

Reviewer #2: The study was very well planned and conducted. I have some minor aspects to comment about.

Methodological aspects:

Would you consider including information about the species social life in the introduction? It has a methodological relevance. I guess they live in isolation; if animals live in pairs, how would you know who reacted to the broadcast?

Response: Thank you for this suggestion. Red squirrels are solitary with the exception of the pup-rearing phase when care is maternal. Thus, distinguishing sex of the respondent is not critical in the context of this study. We have added “Red squirrels maintain solitary territories that they defend against other squirrels regardless of their sex [12]; the only exception to this occurs during parental care when females care for pups from birth through juvenile independence [14].” to the last paragraph of the Introduction.

Line 117-118 you established sampling points at least 300 m apart to avoid detecting the same individual at adjacent locations based in [13,26]. But [13] found that "prior to settlement, offspring made forays of up to 900 m off the natal territory." and "the farthest settlement distance was only 323 m from the natal territory." For that I understand that a young individual in the dispersal phase could eventually be counted twice. Would they also react to broadcasts? Could you distinguish young and adult animals?

Response: A concern when conducting surveys to assess population size is that individuals are not counted twice during the same survey period to avoid inflated estimates of population size (e.g., Ralph CJ & JM Scott. 1981. Estimating numbers of Terrestrial Birds. Studies in Avian Biology No. 6. Cooper Ornithological Society.). Individuals moving between survey periods and consequently being counted at different locations in subsequent periods (e.g. seasons or years) do not adversely influence population estimates as population estimates were made for each survey period independently. Our protocol included keeping track of individuals during visits to a specific point count location to ensure that individuals were not double counted within individual point count periods except as described in our Field Methods section. While red squirrels may be capable of moving the distances noted in your comment over the entire dispersal period (multiple days or weeks; Larsen and Boutin 1994), red squirrels have been recorded as travelling about 500 m per hour (Benhamou 1996) and so would have been unlikely to move between adjacent points within the time period of sampling on a given day, nor would they be double counted in this context. The Benhamou (1996) reference has been added to lines 117-118 of the original submission. With regard to the supplementary comments above, we are not aware of any studies that have examined the age at which red squirrels begin to respond to vocalization broadcast and so we cannot comment on this. At least once a noticeably small-bodied individual (presumably a young of the year) was observed during surveys and included in our count for that location.

Line 120-122: how was that reversion of point count order in alternating months? Once you used the numerical crescent order and then the descrescent order? And what happened when surveys could not be conducted due to precipitation or winds, did you go back to survey some days later?

Response: In alternating months, travel along the paths was reversed such that the first point visited one month became the last point to be visited during the next month’s survey. There was only one occasion when adverse conditions arose during a survey, the survey was abandoned and completed in its entirety on a subsequent day. We have added “Point count order of visitation was reversed” to help clarify this sentence.

Minor corrections:

Line 42: delete space between "abundance and distribution" - DONE

Note that some registries in the refence list ends with a "." and others not. – Thank you for noticing this, a period has been added to the end of all citations.

ADDITIONAL CHANGES

We thank the Reviewers for their comments on this manuscript. Note that we have made a small number of additional editorial changes (words or phrases modified) throughout the manuscript which we feel aid the clarity of our presentation.

6. PLOS authors have the option to publish the peer review history of their article (what does this mean?). If published, this will include your full peer review and any attached files.

Do you want your identity to be public for this peer review? For information about this choice, including consent withdrawal, please see our Privacy Policy.

Reviewer #1: No

Reviewer #2: Yes: Patrícia F. Monticelli

---

## [Editor Report · Decision Letter 1]

31 Jan 2025

Broadcasting American red squirrel vocalizations influences detection probability

PONE-D-24-56431R1

Dear Dr. Warkentin,

We’re pleased to inform you that your manuscript has been judged scientifically suitable for publication and will be formally accepted for publication once it meets all outstanding technical requirements.

Kind regards,

Gisele Akemi Oda, Ph.D.

Academic Editor

PLOS ONE

---

## [Editor Report · Acceptance letter]

PONE-D-24-56431R1

PLOS ONE

Dear Dr. Warkentin,

I'm pleased to inform you that your manuscript has been deemed suitable for publication in PLOS ONE. Congratulations! Your manuscript is now being handed over to our production team.

Kind regards,

on behalf of

Dr. Gisele Akemi Oda

Academic Editor

PLOS ONE